# Novel Carbon Ion and Proton Partial Irradiation of Recurrent Unresectable Bulky Tumors (Particle-PATHY): Early Indication of Effectiveness and Safety

**DOI:** 10.3390/cancers14092232

**Published:** 2022-04-29

**Authors:** Slavisa Tubin, Piero Fossati, Antonio Carlino, Giovanna Martino, Joanna Gora, Markus Stock, Eugen Hug

**Affiliations:** Medaustron Center for Ion Therapy and Research, Radiation Oncology Department, Marie Curie-Straße 5, 2700 Wiener Neustadt, Austria; piero.fossati@medaustron.at (P.F.); antonio.carlino@medaustron.at (A.C.); giovanna.martino@medaustron.at (G.M.); joanna.gora@medaustron.at (J.G.); markus.stock@medaustron.at (M.S.); eugen.hug@medaustron.at (E.H.)

**Keywords:** partial tumor irradiation, particles, carbon ions, bystander effect, abscopal effect

## Abstract

**Simple Summary:**

We present the early results of a novel partial bulky-tumor irradiation using carbon ions and protons. This approach explores the radiation-induced bystander and abscopal effects in patients with recurrent unresectable bulky tumors who failed previous state-of-the-art treatments including radio-chemotherapy. Early tumor response, symptom relief and minimal toxicity suggest that the partial bulky-tumor irradiation was an effective, safe and well tolerated treatment. Abscopal effects elucidate an immunogenic pathway contribution.

**Abstract:**

Background: We present the early results of a novel partial bulky-tumor irradiation using particles for patients with recurrent unresectable bulky tumors who failed previous state-of-the-art treatments. Methods: First, eleven consecutive patients were treated from March 2020 until December 2021. The targeted Bystander Tumor Volume (BTV) was created by subtracting 1 cm from Gross Tumor Volume (GTV) surface. It reflected approximately 30% of the central GTV volume and was irradiated with 30–45 Gy RBE (Relative Biological Effectiveness) in three consecutive fractions. The Peritumoral Immune Microenvironment (PIM) surrounding the GTV, containing nearby tissues, blood-lymphatic vessels and lymph nodes, was considered an organ at risk (OAR) and protected by highly conservative constraints. Results: With the median follow up of 6.3 months, overall survival was 64% with a median survival of 6.7 months; 46% of patients were progression-free. The average tumor volume regression was 61% from the initial size. The symptom control rate was 91%, with an average increase of the Karnofsky Index of 20%. The abscopal effect has been observed in 60% of patients. Conclusions: Partial bulky-tumor irradiation is an effective, safe and well tolerated treatment for patients with unresectable recurrent bulky disease. Abscopal effects elucidate an immunogenic pathway contribution. Extensive tumor shrinkage in some patients might permit definitive treatment—otherwise previously impossible.

## 1. Introduction

The last decades have witnessed remarkable technological, physical and biological developments in the field of radiation oncology resulting in improved treatment effectiveness and reduced complication rates. High-precision tumor identification and localization, as well as dose delivery, permits safe dose escalation, in combination with smart drugs and immunotherapies. However, the resulting improvements in local control and survival do not apply to all oncologic patients. A large patient population is still lacking the benefit of those modern radio-oncology therapies remaining hopeless and predestined to palliative or best supportive care (BSC). These are patients affected by unresectable bulky tumors, whose clinical status is often further complicated by the presence of a large hypoxic tumor component that is resistant to all available non-surgical approaches, (oligo) metastases and poor general conditions, the presence of which only eases the recommendation of the tumor boards for BSC. This is understandable given the large volume of these tumors and their close relationship with neighboring critical organs, which do not allow the ablative radiation dose to be delivered through conventional radiotherapy. Furthermore, the inside of the tumors, especially those large and solid, is represented by a very specific microenvironment that differs a lot from the outer tumor portion in terms of the vasculature, oxygenation level and metabolic profile. The larger the tumor is, the more pronounced these differences are, leading to chronic tumor hypoxia and subsequently resulting in biologically more aggressive tumor behavior, the formation of an immunosuppressive peri-tumoral environment, radio- and chemo-resistance, treatment failure, disease progression and poor prognosis [1,2]. Conventional or low linear energy transfer (LET) radiotherapy is less effective where tumor hypoxia is concerned. From that, the need to improve treatment outcomes for this tumor population arises. Reoxygenation of tumor cells and their position in the cell cycle constitute the rationale for fractionation in photon radiotherapy. However, the impact of this effect on tumor cell killing is significantly reduced by applying heavy particles. The “oxygen effect” on radiation treatment outcome is greater for low-LET radiation, such as photons than for high-LET radiation, such as carbon ions [3,4]. In other words, the challenge posed by tumor hypoxia in large tumors, its associated “radioresistance” and other negative aspects can be addressed more effectively by carbon ion therapy.

In 2016, a novel, unconventional radiotherapy approach for high-dose PArtial Tumor irradiation targeting HYpoxic segment (PATHY) was developed to improve the therapeutic ratio of these large unresectable tumors by exploiting the bystander and abscopal effects [5]. The bystander and abscopal effects are defined as the radiation induced non-targeted anti-tumor effects in terms of tumor cell killing outside the radiation field, thus among those cells that are not irradiated [6]. The bystander effect concerns the death of non-irradiated tumor cells that are in physical contact with those that were irradiated due to the cytotoxic cytokine release from irradiated tumor cells, while the abscopal effect means distant (metastatic) tumor cell killing by the immune system activation following irradiation. The PATHY technique was purposefully designed to add the non-targeted tumor cell killing component to the radiation-mediated tumor cell killing [7,8]. The principle of the PATHY approach consists of delivering a high enough heterogeneously prescribed radiation dose to the hypoxic tumor segment at very precise timing considering the homeostatic immune activity oscillations and sparing the peritumoral immune microenvironment (PIM) as an organ at risk (OAR). Dealing with highly complex clinical scenarios encompassing the bulky tumors that are unresectable and unsuitable for conventional radio-chemotherapy in patients with high symptom burden and low performance status, is an alternative approach to the standard palliative or BSC. It showed encouraging results in terms of immunogenicity and palliation demonstrating that those results are likely caused by immuno-modulatory effects which were previously explained by immunohistochemistry and gene-expression analyses and by the high rate of bystander and abscopal effects. It has been well established that radiation interacts with the tumor and immune system in a number of different ways, being capable of modulating the tumor microenvironment, and affecting innate and adaptive immune systems ideally resulting in abscopal effect or immune-mediated non-targeted distant tumor regression. Accordingly, radiation generates damage-associated molecular patterns leading to increased antigen presentation, priming of antigen-specific naive lymphocytes and finally the production of antitumor T cells [9]. These newly discovered mechanisms are currently exploited by combining radiotherapy and immunotherapy to improve the treatment outcomes [10]. However, the bystander effect is a phenomenon generally observed in animal or cell line experiments and its potential is still not clinically sufficiently exploited [11]. It requires the use of unconventional radiotherapy techniques based on partial tumor irradiation as for example used in GRID or LATTICE whose radiobiology implicates the immunological bystander effects that might lead to superior tumor response when these techniques are employed [12,13]. A strong bystander effect has been demonstrated in unirradiated tumor cells near cells exposed to high dose radiation within the same tumor [14]. The same phenomenon was not reproducible using conventional radiotherapy. Those bystander effects are mediated by cytokines, such as TNF-α and TRAIL [15,16].

In 2020, carbon ions and protons were integrated into the PATHY concept (Particle-PATHY). Their specific physical and biological features make them ideally suited to achieve the goals required more efficiently, in terms of immunomodulatory properties, such as the neutralization of tumor hypoxia, known as a strong immunosuppressor, as well as sparing the outer circumferences of tumor and normal tissue, i.e., the PIM-volume (Peritumoral Immune Microenvironment). Compared to photons, particles can significantly reduce the “integral immune dose” sparing more efficiently PIM. Furthermore, carbon ions induce significantly stronger immunogenic cell death than photons or protons in both normoxic and hypoxic conditions [17,18,19]. In contrast, both photons and protons are challenged to accomplish complete tumor cell kill in hypoxic conditions. [20]. In addition, carbon ions are heavy charged particles with high LET. Their higher number of ionization events along the beam path may have superior biological advantages in terms of immunogenicity involving different cell death pathways [21].

We conducted a retrospective review approved by the local ethic committee to assess the impact of Particle-PATHY on Quality of Life (QoL), tumor downsizing (neoadjuvant effect), tumor control and survival among patients affected by recurrent unresectable bulky tumors unsuitable for conventional radio-chemotherapy. This series includes the first eleven consecutive patients treated at our institution from March 2020 until December 2021 that were previously subjected to BSC and who were referred for Particle Therapy but considered unsuitable for conventional Particle Therapy by our New Patient Review Board (Figure 1).

The hypothesis is that for effective immunomodulation manifested by a clinically relevant bystander and abscopal effects, the entire tumor volume may not need to be irradiated but only a partial volume to initiate the immune cycle in radiation-spared PIM.

## 2. Materials and Methods

***Simulation and immobilization*:** CT and MRI scans were performed with 2 mm slice thickness as per institutional practice and were subsequently co-registered at the level of the bulky tumor. Depending on the tumor site, patients were positioned prone or supine and immobilized using a thermoplastic mask and customized vacuum cushion. Simulation CT was performed without whilst the MRI with contrast media. The MRI protocol consisted of T1ce, T2w (including FLAIR and MutliVane techniques) and DWI sequences.

***Volumes definition*:** The targeted internal partial tumor volume is termed “Bystander Tumor Volume” (BTV). Its delineation was adopted from previously published studies on photon-based SBRT-PATHY [7,8]. Since in those series the BTV corresponded on average to the centrally located approximately 30% of Gross Tumor Volume (GTV), in this study it was created by subtracting 1cm from the GTV surface in all directions until reaching 30% (+/−3%) of the GTV volume (Figure 2). No additional margins, neither for the Clinical Target Volume (CTV) nor for Planning Target Volume (PTV), were applied to the BTV since the objective of this treatment was to irradiate a tumor sub-volume in order to effectively spare the peritumoral immune system cells. No hypoxia-specific tumor imaging, which is planned for future patients, was utilized in this present group. However, the majority, if not all, of the hypoxic tumor segments were likely covered by the BTV volume since in all cases the central, necrotic and perinecrotic tissues were included in the BTV.

Once the BTV was defined, the PIM was created (Figure 2A,B). The PIM, containing the loco-regional infiltrating immune system cells as the mediators of the non-targeted radiation effects, was considered an OAR (Organ At Risk). The PIM was created by adding a uniform 1 cm margin to the GTV and then subtracting the same GTV to create “a ring” that surrounds the GTV. The goal of treatment planning was to keep the dose within the PIM as low as reasonably achievable but considering the following dose constraints as the planning objective: Dmean < 6 Gy, D20 < 0.5 Gy, D30 < 1 Gy and D50 < 3 Gy in three fractions. Additionally, all nearby regional vascular supply and lymph node stations were included if they were within the 2 cm-distant areas from the GTV.

The OARs at the level of the treated area were delineated on MRT images and subsequently checked and refined on the co-registered simulation CT images. Those included the structures involved by or in close proximity to the tumor, such as parenchymatous and hollow organs, vessels and nerves.

***Treatment planning*:** Treatments were prescribed using either protons, carbon ions, or mixed proton-carbon treatment. Comparative plans (carbon ions vs protons) were performed for each patient. The choice of the particle-type was subordinated to *the BTV-coverage*: *PIM-dose ratio.* The goal was to maximize target coverage while minimizing dose to PIM. Radiotherapy plans were calculated on the native simulation CT scans using Ray Station 8 and 11 with Pencil Beam (for carbon ions) and Monte Carlo (for protons) algorithm.

***Dose prescription*:** The task of prescribing the PATHY dose addressed two separate dosimetric issues: concerning immunogenicity aiming to increase the likelihood of generating immune-mediated bystander and abscopal effects:(1)To deliver sufficiently high and possibly heterogeneously distributed radiation dose at the level of the targeted BTV. The intended dose was in the order of 10–15 Gy per fraction. In order to address heterogeneity in radiation dose delivery, dose-gradients were created within the BTV, by creating an additional BTV (BTV2) within the former BTV (BTV1) by subtracting 5 mm from the BTV1 surface in all directions whenever tumor volume would permit it (Figure 2B and Figure 3B).(2)To ascertain simultaneously effective radiation sparing of the PIM in order to preserve the loco-regional immune system cells necessary to mediate the non-targeted radiation effects.


Achieving the BTV prescription goal was subordinated to the possibility of creating a sufficiently sharp dose fall off between the BTV surface and the GTV surface surrounded by the PIM and was proportional to the thickness of peripheral tumor tissue in between the BTV and GTV surfaces. Treatment plans were calculated to deliver 30–36 Gy in three consecutive fractions (10–12 Gy per fraction) to the BTV1, and 36–45 Gy (12–15 Gy per fraction) to the BTV2 so that at least 95% of the BTVs received as minimum 95% of the prescription dose. The choice of prescribing the fractional doses was driven by PIM-sparing: 12 Gy to the BTV1 with 15 Gy to the BTV2 were preferred if allowing planned PIM-sparing which was easier to achieve in particularly large tumors. Otherwise, the fractional dose was reduced to 10 Gy to the BTV1 with 12 Gy to the BTV2 to lower the dose received by PIM. In those patients in whom neither of the two goals was possible, only one BTV (BTV1) was created and treated either with 12 Gy × 3 or 10 Gy × 3 (Figure 2A and Figure 3A). Reported doses in Gy are RBE-weighted doses calculated from the physical dose using the LEM model for carbon ions [22] while for protons a fixed RBE of 1.1 was assumed.

***Follow-up*:** The RECIST-based [23] assessment of treatment response was performed at 4 weeks after treatment by using CT and MRI, followed by repeated scans at 8 weeks and then every 3 months. Toxicity was evaluated using the CTCAE Criteria [24].

All procedures performed in the present study were in accordance with the ethical standards. All patients signed informed consent.

## 3. Results

### 3.1. Patient and Disease Characteristics

Patient and disease characteristics are summarized in Table 1 and Table 2. There were seven men (64%) and four women (36%). The average patient age was 56 years (range 25–78). Most of the patients were in poor general conditions, the median KPS was 60% with >50% of the patients´ KPS as low as 30–60%. All patients had typically large, symptomatic, recurrent bulky tumors (stage rcT4) that, due to the very high volumes and intimate relationship with nearby critical structures, did not have any treatment option including conventional radiotherapy or particle therapy. They were all pretreated per standard of care treatments, including systemic therapy, surgery, and radiotherapy resulting subsequently in progressive disease (PD). Particularly, five patients (45%) had prior photon radiotherapy at the local site, two of whom already received re-irradiation before Particle-PATHY. The average previous radiation dose was 82.4 Gy (range 60–120). Five patients (45%) had been administered chemotherapy, biologic, or immunotherapy without response. Ten out of eleven patients (91%) had previously undergone surgery, of which eight (73%) were treated with multiple surgeries. Only two patients (18%) received subsequent immunotherapy due to PD following Particle-PATHY, without response. Five patients (45%) had advanced-regional (cN3) or distantly metastatic (cM1) disease allowing assessment for abscopal effect. All patients had significant tumor-related symptoms that were uncontrolled prior to treatment. Seven out of eleven (63%) patients had sarcoma.

### 3.2. Treatment Characteristics

Treatment characteristics are summarized in Table 3. Irradiated bulky-tumor sites were skull base (18%), head and neck (18%), lung (9%), abdomen (27%) and pelvis (27%). All three treated abdominal lesions were intraperitoneal. Mean tumor diameter and volume were 15.6 cm (range 6–27.5) and 1460 cc (range 76–5645.9), respectively. The mean volume of targeted BTV was 431.7 cc (range 36.1–1494) representing on average 29.6% of the mean GTV volume; i.e., approximately 1/3 of the GTV was targeted. Six patients (54%) were treated with 10 or 12 Gy delivered three times on consecutive days to the single BTV volume (BTV1). However, in the remaining five patients (46%) affected by particularly large tumors the dose corresponding to 10 and 12 Gy or 12 and 15 Gy was delivered three times to two tumor sub-volumes (BTV1 and BTV2), respectively (Figure 2 and Figure 3). Seven patients (64%) were treated with mixed proton and carbon ion therapy while four of them either with full proton (18%) or carbon ion (18%) therapy. The “immune-dose” in terms of the PIM-sparing was as low as it was initially intended: the total average mean dose was 5.8 Gy, D50 ≤ 2.8 Gy, D30 ≤ 0.7 Gy and D20 ≤ 0.2 Gy in three fractions.

### 3.3. Clinical Outcomes

Clinical outcomes are summarized in Table 4 and Figure 4 and Figure 5. The median follow up was 6.3 months with a range of 3–16 months. Local control (LC) at 3 months had been achieved in 8/11 patients (73%) (Figure 5). All three patients with tumor progression (PD) within the first three months had previously extremely fast growing tumors (two chondrosarcomas and one non-seminomatous germ cell tumor). At the last follow-up, six patients (54%) experienced local PD (in addition to the previously described three early recurrences, one skull base sarcoma, one squamous oral cavity and one skull base chordoma). Five patients (46%) maintained local responses. Among these, one patient (9%) had a complete response (CR) (adenocarcinoma of nasal cavity) and another one had an almost-CR (soft-tissue sarcoma that is still regressing). In terms of the neoadjuvant potential, significant tumor-downsizing had been achieved in eight patients (73%) (Figure 6, Figure 7 and Figure 8). It was not possible to determine any statistically significant difference in terms of the local tumor control between patients treated with two BTVs and a higher radiation dose compared to those treated with a single BTV and a lower radiation dose. Likely, a larger sample of patients in both groups was required to detect any anti-tumor effect differences due to the dose-size and heterogeneity.

The neoadjuvant effect was not induced within the short follow-up time by those typically late- and scarce responding chordomas and chondrosarcomas, as expected. The average tumor-volume regression was 890.4 cc corresponding to 61% of the mean GTV volume. LC had an average duration of 5 months (range 2–16).

Overall survival was 64% with a median survival of 6.7 months (range 3–16). 46% of patients were progression-free with a median progression-free survival (PFS) of 5.2 months (range 2–16).

Symptom relief was achieved in 10/11 patients (symptom control rate 91%). KPS was significantly improved in 10 (91%) patients by an average of 20% (range 10–60%; 10%-improvement achieved in a patient with prior 90% KPS). Three patients (27%) experienced Grade 1 fatigue. No other side effects were observed. In ten patients (91%) treatment resulted in a reduction of analgesics and abrogation of strong opioids.

The abscopal effect has been observed in three (60%) out of five patients that had assessable distant tumor disease either N+ or M+ (Figure 6, Figure 7, Figure 8 and Figure 9). These were patients 4, 6 and 7 (Figure 1, Table 2). Patient 6 had a single 2.4 cm large synchronous peritoneal implant metastasis. Patients 7 and 4 had multiple synchronous metastases: at the level of the peritoneum (2.3 and 1.6 cm), and involving regional neck lymph nodes (the largest 6 cm), respectively (Figure 6, Figure 7 and Figure 8). In the case of patients with multiple metastases (i.e., in patients 4 and 7) all metastases demonstrated an abscopal effect by tumor volume reduction. Average metastases-volume regression was 42% (range 36–48). The average abscopal response lasted as long as the local response of the partially treated tumors, which in the case of these three patients was 4 months. While metastases that had previously decreased due to the abscopal effect in patients 4 and 7 progressed after 5 and 2 months, respectively, the single metastasis in patient 6 was still regressing as well as his partially irradiated primary tumor at the last follow up at 5 months.

Table 4 shows the clinical outcomes, excluding from data analysis the four patients affected by those typically late-responding and scarce-shrinking chordomas and chondrosarcomas, due to their very specific architecture, mainly consisting of extracellular matrix with little cells and stroma, which is significantly different from those “conventionally” solid tumors that are very rich in cellular components.

## 4. Discussion

This analysis presents the early results of a novel Particle-PATHY concept using high-dose, hypofractionated carbon ions and protons for patients with recurrent unresectable bulky tumors who failed previous state-of-the-art treatments including radio-chemotherapy. This analysis focuses on the initial response and early outcomes for patients with progressive disease and without any realistic standard therapeutic option, thus for whom even a short follow-up period may allow decent conclusions to be drawn. By delivering this short-course treatment, we were able to improve KPS and QoL, prolong survival, induce substantial tumor downsizing, effectively reduce severe symptoms and suspend or reduce the use of opioids (Table 4). Treatment was well tolerated and associated with negligible acute and early toxicity. It is delivered in an overall short time frame of several days only—which is favored by both patients and physicians. The clinically unequivocal benefit was obtained in a patient population who did not previously have any option other than BSC, and whose predicted life expectancy according to Palliative Prognostic Index [25] did not exceed two months in 72% of patients, if left untreated. Indeed, available published data for this patient group indicates an estimated median overall survival time of 25–54 days only [26,27,28,29].

The majority of tumors progressed rapidly locally prior to initiation of PATHY-therapy, i.e., with a radiographic increase in size. This demonstrated the need for a fast and effective therapeutic response. In such large tumors with fast growth dynamics, any anti-tumor effect of conventional palliative radiotherapy would probably be poor requiring several weeks before reaching its maximum—thus frequently exceeding the estimated overall survival time of the patient. In our series, 64% of patients are still alive reaching 6.7 months (range 3–16) of median survival and 5.2 months (range 2–16) of median PFS. Additionally, LC was maintained on average for 5.4 months. This 3-days short-course treatment shows a favorable cost-effectiveness profile by minimizing requirements and efficient use of resources, i.e., equipment, treatment rooms and staff.

The scope of this study was not to assess the underlying physiologic mechanisms resulting in favorable clinical outcomes. However, previous clinical and biomolecular data provided evidence of interaction with the immune system [7]. Those published results, characterized by high bystander- and abscopal-response rates, combined with immunohistochemical and gene-expression analyses of partially irradiated bulky- and unirradiated abscopal-tumor sites, suggest that delivery of the high-dose radiation to the partial tumor volume with PIM-sparing significantly boosts the immune-mediated anti-tumor effects. Present study data confirm the immunogenic potential of the PATHY approach since 3/5 of patients (60%) with metastatic, unirradiated disease demonstrated abscopal effects (Figure 6, Figure 7 and Figure 8). Irradiation can result in several immunogenic anti-tumor effects including increased generation and presentation of neo-antigens with subsequent maturation and activation of antigen presenting cells, priming of lymphocytes, and release of immuno-stimulatory cytokines, increased CD-8 T-cell tumor-infiltration, finally leading to immunogenic cell death [30,31]. The clinical contribution of induction of abscopal effects is scarce following the use of conventional radiotherapy. This is possibly related to the physical dose distribution of photon-based radiotherapy, resulting in unintentional significant radiation dose to nearby immune systems cells and organs. Those need to be spared in order to permit immuno-stimulation. Hence, conventional (“whole-tumor”) photon-based radiotherapy is not optimally designed to be immuno-simulative. This possibly explains that abscopal effects are rarely reported following conventional photon-based radiotherapy—taking also into consideration the reports of the correlation between radiation-induced lymphopenia and poor oncologic outcome and survival [32,33,34].

Particle-PATHY was delivered in this patient group with strictly palliative intent. In tumors exhibiting good response, i.e., extensive tumor shrinkage, the issue arises, if downsizing might permit more definitive and potentially even curative treatment. This would be in analogy to the use of pre-operative irradiation to convert borderline resectable or locally advanced unresectable bulky tumors to a resectable state. Radiotherapy has been used for decades as a neoadjuvant treatment aiming to downsize large tumors allowing for radical resection [35,36,37]. We observed a significant downsizing in 73% of complex tumors with an average tumor-volume reduction of 61% (Table 4). Excluding the late-responding and scarce-shrinking chordomas/chondrosarcomas with inherent matrix architecture, the neoadjuvant effect among solid tumors was 100% (Table 4).

Particle-PATHY delivered extraordinarily low radiation dose to the surrounding OARs–best reflected by the low PIM-dose (Table 3). Since surrounding tissues and OARs receive low dosages only, Particle-PATHY will permit integration of and combination with various definitive treatments—in case PATHY accomplishes significant downsizing. This might include surgery, chemotherapy, or radio-chemotherapy. In our series, Particle-PATHY therapy was administered only once. If indicated, it can be delivered potentially several times in order to maximize its effectiveness due to the low dose to surrounding structures.

This study represents a first and early analysis. Hence, it has inherent limitations. The presently small patient numbers and short follow-up period preclude a more comprehensive statistical analysis. Additional follow-up is presently accumulating. We considered these retrospective data sufficiently promising to initiate a prospective, IRB-approved Study.

## 5. Conclusions

The Particle-PATHY approach resulted in an effective, safe and well tolerated treatment.

Our early data indicate that as a minimum, all patients gained improvement in their symptoms and quality of life without associated treatment related toxicity. The local tumor regrew in some patients within the follow-up period. However, in a significant amount of patients Particle-PATHY resulted in varying degrees of tumor downsizing. This holds the true prospect that in some patients, for whom no standard-of-care definitive treatment existed, Particle-PATHY can result in a shrinkage of tumor size that will permit the application of definitive treatment thereafter. This prospect will be explored in the future. The fact that abscopal effects in un-irradiated distant metastatic sites were observed in 3/5 patients points towards the involvement of immune-stimulating mechanisms.

Optimum patient selection and definition of most suitable disease characteristics are currently explored in an ongoing, prospective study. The added benefit of particle therapy compared to photon RT will ultimately require randomized, multi-institutional studies.

## 6. Patents

Tubin Slavisa M.D. reported an international patent application PCT/EP2019/052164 published as WO 2019/162050. The authors reported no other conflict of interest.

## Figures and Tables

**Figure 1 cancers-14-02232-f001:**
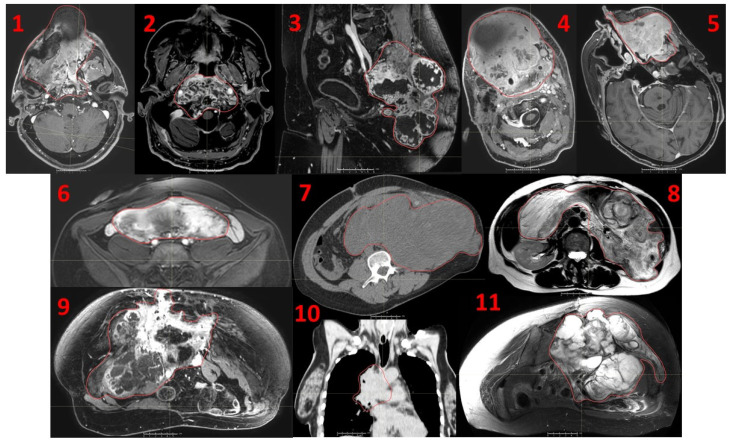
Eleven consecutive patients affected by recurrent unresectable bulky tumors treated with a novel Particle-PATHY approach: 1. Leiomyosarcoma, 2. Skull base chordoma, 3. Sacral chordoma, 4. Squamous-cell carcinoma of oral cavity, 5. Adenocarcinoma of nasal cavity, 6. Intraperitoneal soft-tissue sarcoma, 7. Intraperitoneal non-seminomatous germ-cell carcinoma, 8. Intraperitoneal liposarcoma, 9. Pelvic chondrosarcoma, 10. Squamous-cell lung carcinoma, 11. Pelvic chondrosarcoma. Red contours indicate the Gross Tumor Volume (GTV).

**Figure 2 cancers-14-02232-f002:**
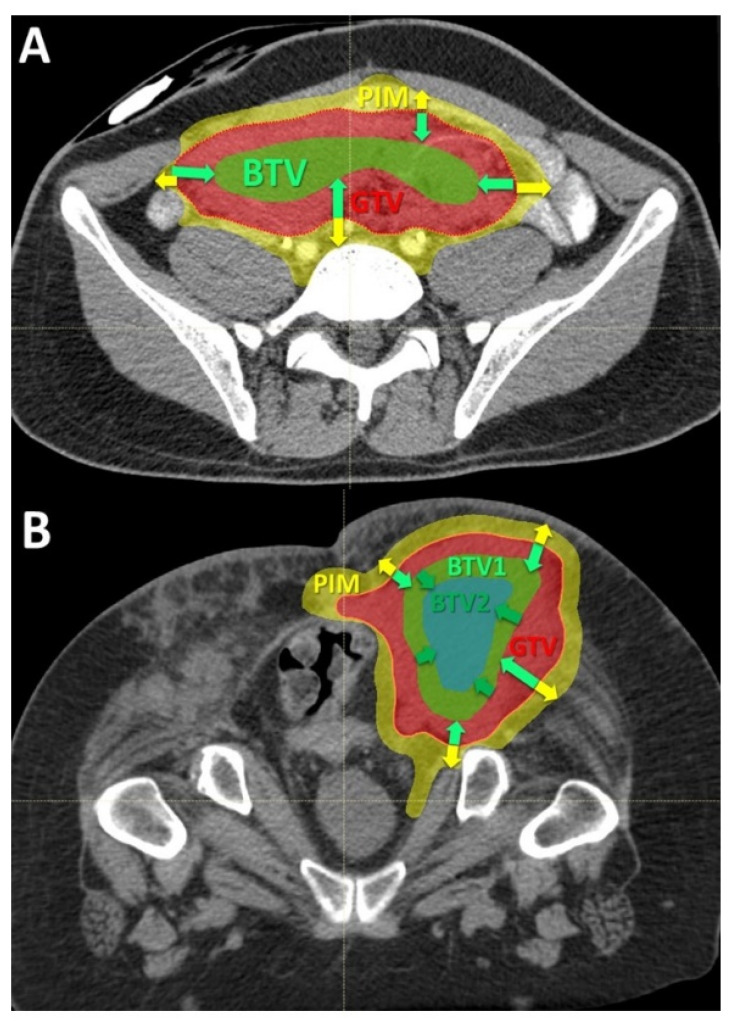
Delineation of the Bystander Tumor Volume (BTV) and Peritumoral Immune-Microenvironment (PIM): (**A**) the figure shows the principles of the target delineation for Particle-PATHY treatment. After the Gross Tumor Volume (GTV, red dotted line) has been defined, the BTV-partial tumor volume (green line) as the target for PATHY irradiation has been created by subtracting 1 cm from GTV surface in all directions reducing the GTV volume to the 30% (+/−3%). PIM (yellow line) was created by adding a uniform 1 cm margin to the GTV and then subtracting the same GTV to create “a ring” that surrounds the GTV; (**B**) In those patients affected by particularly large tumors two tumor sub-volumes BTV1 and BTV2, respectively, were delineated whereby the BTV2 was created by subtracting 1 cm from BTV1 (or 2 cm from GTV surface) in all directions.

**Figure 3 cancers-14-02232-f003:**
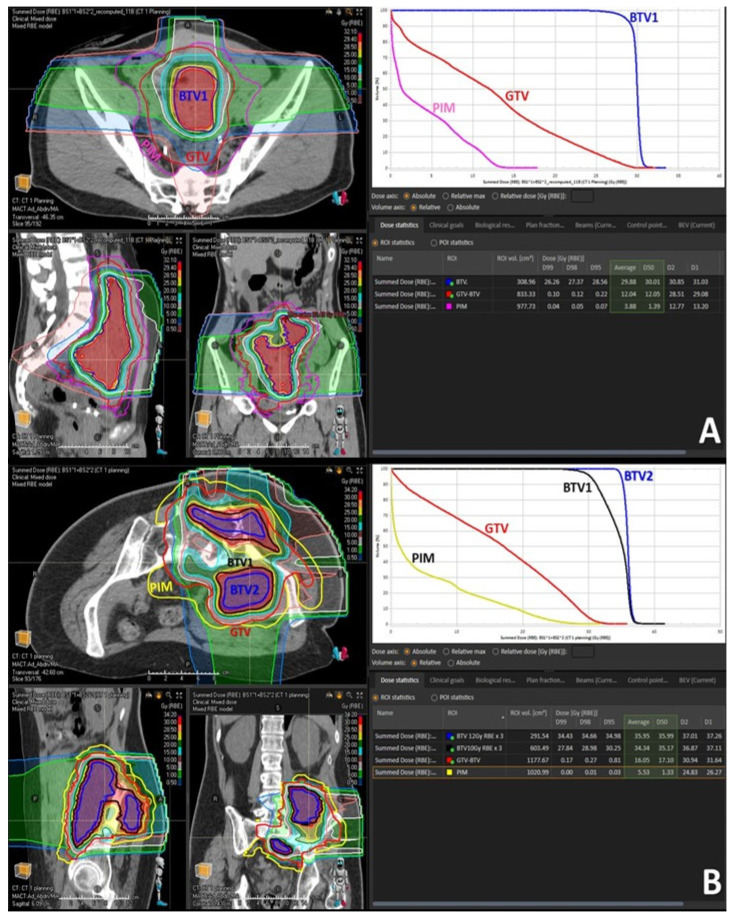
Treatment volumes for particle-based partial bulky tumor irradiation: the figure shows the targeted tumor sub-volumes with corresponding dose distribution delivered to the single subvolume-BTV1 (**A**) or two sub-volumes, BTV1 and BTV2, respectively (**B**). The highlighted green area in the dose statistics indicates the average and 50% dose of the targeted BTV volumes, non-targeted GTV volume and radiation-spared PIM volume. It is worth mentioning here that the mean radiation dose at the level of non-target GTV (GTV-BTV) was radio-biologically very low, hence non-tumoricidal, corresponding in the first case (**A**) to only BED2 Gy = 14 Gy while in the second case (**B**) to only BED2 Gy = 20 Gy.

**Figure 4 cancers-14-02232-f004:**
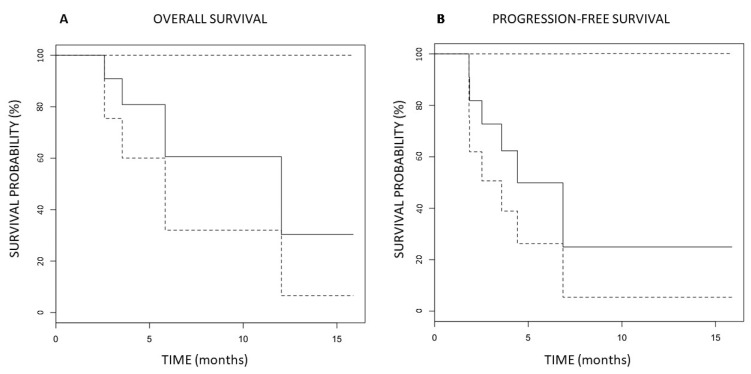
Actuarial Kaplan-Meier overall (**A**) and progression-free survival (**B**) of 11 patients affected by unresectable recurrent bulky tumors treated with PARTICLE-PATHY. The dashed lines represent the 95% confidence intervals.

**Figure 5 cancers-14-02232-f005:**
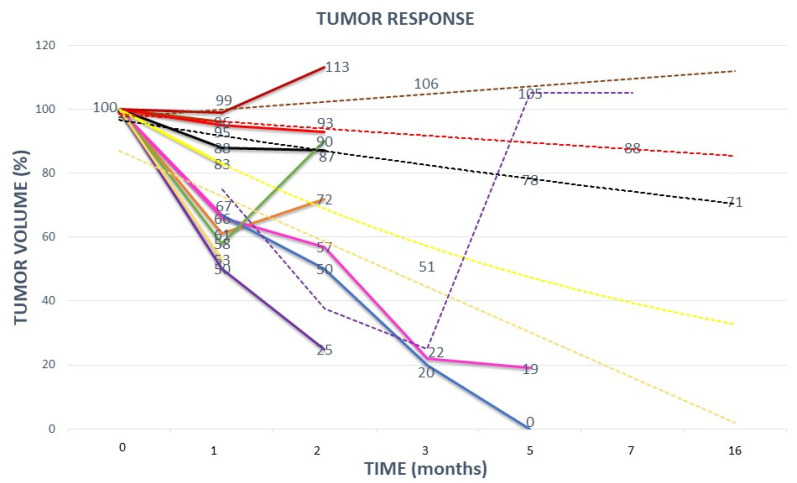
TUMOR RESPONSE: This graph shows the tumor response to partial radiation from each of the eleven treated patients. Absolute tumor volumes were translated into relative volumes for more clear presentation of results (differences between tumor volumes were too large leading to a suboptimal graph-presentation if absolute tumor volumes were used). Dashed lines represent the trend of tumor response in those patients in whom data on tumor response were lacking due to certain unexpected logistic issues (for example: a patient due to COVID-19 infection was prevented from performing a control MRI). The ordinal numbers of the eleven listed patients refer to Figure 1: 1. orange line; 2. red line; 3. yellow line; 4. purple line; 5. blue line; 6. pink line; 7. green line; 8. black line; 9. brown line; 10. gold line; 11. dark-red line.

**Figure 6 cancers-14-02232-f006:**
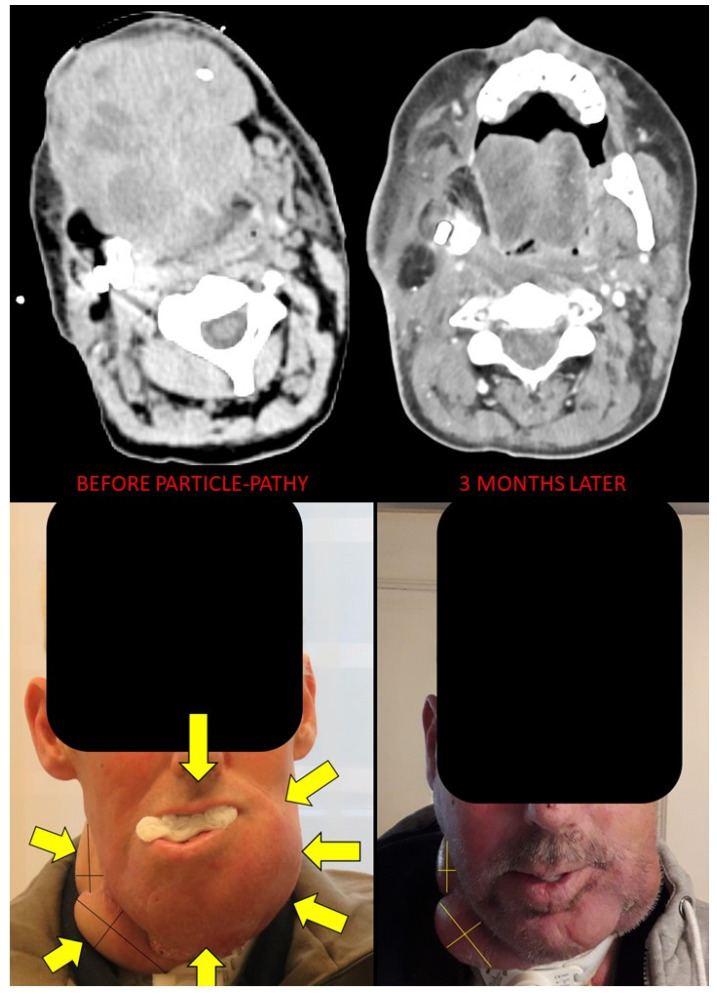
Bystander and abscopal effects generated by Particle-PATHY: figure shows the almost complete disappearance of partially irradiated recurrent squamous cell bulky tumor of the oral cavity due to the bystander effect and regression of unirradiated right neck lymph node metastases due to the abscopal effect three months following Particle-PATHY (left: before Particle-PATHY; right: after). This patient was previously already irradiated with 70 Gy external beam radiotherapy and could not be re-irradiated either with conventional photon nor with particle therapy.

**Figure 7 cancers-14-02232-f007:**
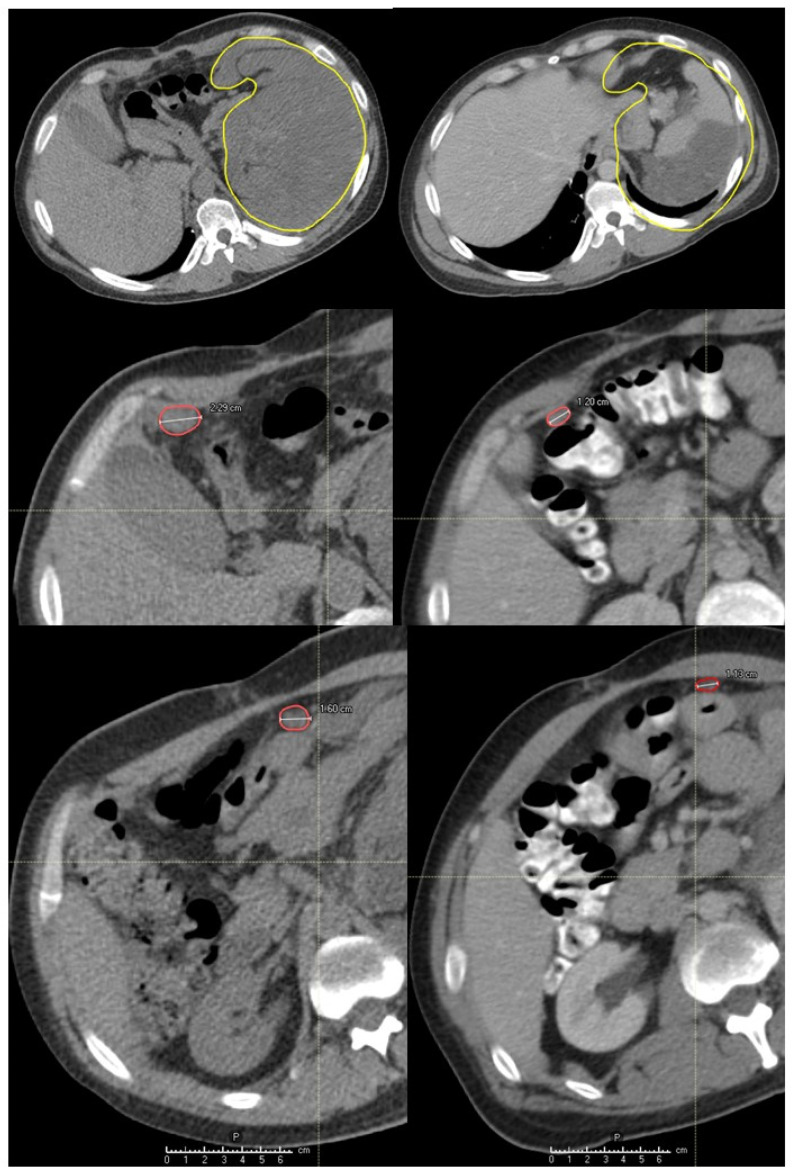
Bystander and abscopal effects generated by Particle-PATHY: the figure shows a significant regression of partially irradiated recurrent intraperitoneal non-seminomatous germ cell bulky tumor (yellow contour) due to the bystander effect and regression of unirradiated regional metastases (red contours) due to the abscopal effect one month following Particle-PATHY (left: before Particle-PATHY; right: after). It was not possible to offer to this 33-year-old patient any kind of conventional radiotherapy including particle therapy.

**Figure 8 cancers-14-02232-f008:**
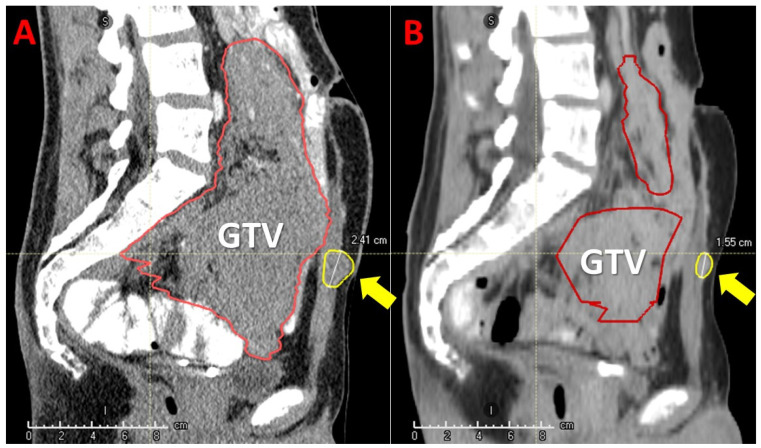
Bystander and abscopal effects induced by Particle-PATHY: the figure shows an aggressive unresectable intraperitoneal recurrent bulky desmoid tumor in a 24-year-old patient previously submitted to surgery twice and treated with immunotherapy with no response. On the left (**A**) tumor extension before Particle-PATHY (Gross Tumor Volume-GTV, red contour) with an additional separate lesion in the abdominal wall (yellow contour). On the right (**B**) 78% tumor-volume reduction (red contour-residual GTV) due to the bystander effect, but also 50% volume-reduction of the untreated separate lesion (yellow contour-residual tumor) due to the abscopal effect 4 months following the Particle-PATHY.

**Figure 9 cancers-14-02232-f009:**
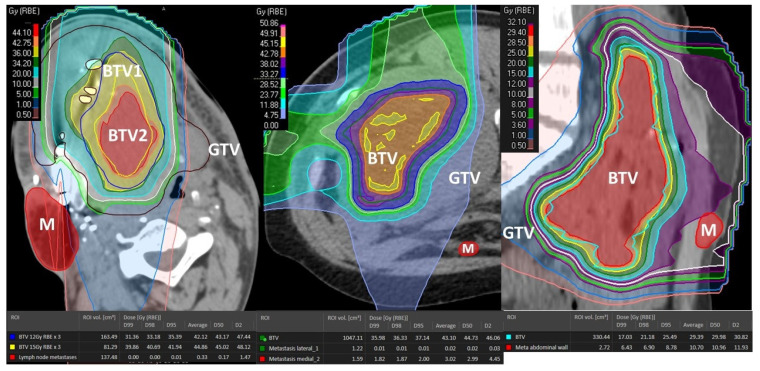
Dose distribution of the abscopal-responding unirradiated metastases. GTV: partially irradiated Gross Tumor Volume; BTV1: Bystander Tumor Volume treated with 12 Gy × 3; BTV2: Bystander Tumor Volume treated with 15 Gy × 3; M: metastasis-abscopal tumor site.

**Table 1 cancers-14-02232-t001:** Patient and disease characteristics.

	Total Patients: 11
**PRIOR COURSES OF RADIOTHERAPY**:	5/45%
Single Course	3/27%
Two Courses	2/18%
Average prior total radiation dose	82.4 Gy (range 60–120)
**HISTOLOGY**:	
Soft-tissue sarcoma	3/27%
Bone sarcoma (chondrosarcoma, chordoma)	4/36%
Adenocarcinoma (H&N)	1/9%
Squamous (H&N, Lung)	2/18%
Non-seminomatous germ cell	1/9%
**TUMOR STAGE**:	**All rcT4**
**rcT4** cN0 cM0	6/55%
**rcT4** **cN3** cM0	1/9%
**rcT4** **cN1-3** **cM1**	4/36%
**TUMOR-RELATED SYMPTOMS**:	**11/100%**
Pain: average VASmax 7	10/91%
Bleeding: average grade 3	3/27%
Dysphagia: average grade 2	3/27%
Two or more symptoms	8/73%

**Table 2 cancers-14-02232-t002:** Selected tumor and treatment characteristics of the eleven patients treated with Particle-PATHY. EBRT-external beam radiotherapy; CHT-chemotherapy; BT-brachytherapy; PBT-proton beam therapy; P-protons; C-carbon ions; CIRT-carbon ion radiotherapy.

Patient	Histology	Tumor Volume (cc)	Prior Surgery (*n*)	Prior Systemic Therapy (*n* of Lines)	Prior RT and/or CHT	Particle-Pathy Dose Gy (RBE)	Follow Up (Months)
1	Leiomyosarcoma	298	3	3	(1.) RT-CHT 70Gy; (2.) EBRT 50 Gy + BT with Ruthenium applicator (dose unknown, 1977).	12 Gy × 3: 1P + 2C	3
2	Chordoma	76	NO	NO	PBT 76Gy	12 Gy × 3: full CIRT	5
3	Chordoma	1102	1	NO	NO	12 Gy × 3 (BTV1), 15 Gy × 3 (BTV2): 1C + 2P	5
4	Squamous-cell carcinoma	435	6	4	RT-CHT 60Gy	12 Gy × 3 (BTV1), 15 Gy × 3 (BTV2): 1C + 2P	6
5	Adenocarcinoma	126	2	NO	RT-CHT 66Gy	12 Gy × 3 (BTV1), 15 Gy × 3 (BTV2): 1C + 2P	5
6	Aggressive desmoid tumor	1150	3	1	NO	10 Gy × 3: 1C + 2P	6
7	Non-seminomatous germ cell carcinoma	5646	3	6	NO	10 Gy × 3: full P	4
8	Atypical mixoid liposarcoma	2868	4	NO	NO	10 Gy × 3: full CIRT	16
9	Chondrosarcoma	2548	3	NO	NO	12 Gy × 3 (BTV1), 15 Gy × 3 (BTV2): full CIRT	10
10	Squamous-cell carcinoma	126	3	4	(1.) RT-CHT 60 Gy; (2.) RT 30 Gy;	10 Gy × 3: full P	4
11	Chondrosarcoma	1795	1	NO	NO	10 Gy × 3 (BTV1), 12 Gy × 3 (BTV2): 1C + 2P	5

**Table 3 cancers-14-02232-t003:** Treatment characteristics.

**Treated Bulky Tumor Site:**	
H&N	2/18%
Lung	1/9%
Abdomen (intraperitoneal!)	3/27%
Pelvis	3/27%
Skull base	2/18%
**BULKY-DIAMETER/VOLUME:**	
Diameter mean/range (cm)	15.6/6–27.5
Volume mean/range (cc)	1460/76–5645.9
**TARGETED BTV-VOLUME:**	
Volume mean/range (cc)	431.7/36.1–1494
Percent of the total bulky tumor-volume	29.6%
**DOSE PRESCRIPTION:**	
10 Gy × 3	4/36%
12 Gy × 3	2/18%
10 and 12 Gy × 3 (BTV1 and BTV2)	1/9%
12 and 15 Gy × 3 (BTV1 and BTV2)	4/36%
**PARTICLE TYPE:**	
Protons	2/18%
Carbon ions	2/18%
Mixed (P + C)	7/64%
**IMMUNE-SPARING: PIM-DOSE**	
Dmean (avg.)	5.8 Gy/3 fractions
D50% (avg.)	≤2.8 Gy/3 fractions
D30% (avg.)	≤0.7 Gy/3 fractions
D20% (avg.)	≤0.2 Gy/3 fractions

**Table 4 cancers-14-02232-t004:** **A.** Clinical outcomes including all 11 patients. **B.** Clinical outcomes excluding tumors unsuitable for PATHY approach because of the very specific tumor architecture (chordomas/chondrosarcomas) characterized by very little cellular and stromal component and excess of extracellular matrix which makes the target for PATHY very difficult to define.

**A**	
**Follow Up (Median/Range)**	**6.3/3–16**
**BULKY-TUMOR CONTROL at 3 months:**	**8/73%**
PR	8/73%
PD	3/27%
**OVERALL BULKY-TUMOR CONTROL**	5/46%
CR	1/9%
PR	4/37%
PD	6/54%
**DOWNSIZING (neoadjuvant effect):**	
achieved	8/73%
not achieved	3/27%
**OVERALL SURVIVAL**	
alive	7/64%
dead	4/36%
Median survival	6.73 months (range 3–16)
**PROGRESSION-FREE SURVIVAL**	
progression-free	5/46%
progressed	6/54%
Median progression-free survival	5.16 months (range 2–16)
**SYMPTOM(s) RELIEF:**	
yes	10/91%
no	1/9%
**KPS IMPROVEMENT after PATHY** (median/range)	**91% (20%/10–60%)**
**SIDE EFFECTS:**	**3/27%**
fatigue	3/27%
others	0/0%
**PAIN KILLERS REDUCTION**	**10/91%**
**ABSCOPAL EFFECT** (5 Pts with assessable N+/M+)	**3/60%**
**TUMOR-VOLUME REGRESSION** avg (cc/%)	**890.4/61%**
**DURATION OF LOCAL CONTROL** (avg/range) (months)	**5.4/2–16**
**B**	
**Bulky-Tumor Control Achieved within 3 Months:**	**7/100%**
PR	7/100%
SD	0/0%
PD	0/0%
**BULKY-TUMOR CONTROL at last FUP:**	
CR	1/14%
PR	3/43%
SD	0/0%
PD	3/43%
**DOWNSIZING (neoadjuvant effect):**	
achieved	7/100%
not achieved	0/0%
**PROGRESSION-FREE SURVIVAL** (avg/range) (months)	**5.9/2–16**
**DURATION OF LOCAL CONTROL** (avg/range) (months)	**5.9/2–16**

## Data Availability

Concerning the Data Availability Statements the interested parties could contact directly the corresponding author.

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
