# Peer review of "Novel Carbon Ion and Proton Partial Irradiation of Recurrent Unresectable Bulky Tumors (Particle-PATHY): Early Indication of Effectiveness and Safety"

_cancers, 2022, doi:10.3390/cancers14092232_

Round 1
Reviewer 1 Report
The authors present first results of a novel partial bulky-tumor irradiation using particles for patients with recurrent unresectable bulky tumors who failed previous state-of-the-art treatments. The hypothesis is that for an effective immunomodulation manifested by clinically relevant bystander and abscopal effects, the entire tumor volume may not need to be irradiated but only a partial volume to initiate the immune cycle.
The authors conclude that partial bulky-tumor irradiation is an effective, safe and well tolerated treatment care for patients with unresectable recurrent bulky disease.
The manuscript is very well written. I have only some minor comments the authors should address in a revised version.
Introduction: The authors write: "Bystander effect, which might be considered as the local abscopal effect is a phenomenon generally observed in animal or cell line experiments." I would not consider bystander effects as local abscopal effects. Bystander effects can be measured also in vitro without the presence of immune cells. The two mechanisms are very different.
Discussion: The authors write that particle-PATHY can result in a shrinkage of tumor size. This is very important , because it may permit the application of definitive treatment thereafter. Could the authors also discuss the possibility of a long term tumor control after particle-PATHY?
Author Response
Thank you very much for all constructive comments and suggestions! We did take them all in consideration, point by point, in order to improve the quality of our paper:
“1.) Introduction: The authors write: "Bystander effect, which might be considered as the local abscopal effect is a phenomenon generally observed in animal or cell line experiments." I would not consider bystander effects as local abscopal effects. Bystander effects can be measured also in vitro without the presence of immune cells. The two mechanisms are very different.”
Response: yes of course. You´re right. What has been written was more symbolically said in order to help readers who are not familiar with the non-targeted effects of radiotherapy easier to understand the point…That statement didn´t take into account the scientific background and the mechanisms of those two phenomena, which was, explained later on in the text…However, we improved that. Thanks!
“2) Discussion: The authors write that particle-PATHY can result in a shrinkage of tumor size. This is very important, because it may permit the application of definitive treatment thereafter. Could the authors also discuss the possibility of a long term tumor control after particle-PATHY?”
Response: Indeed, this is a very important issue. We have already discussed in the text that the observed tumor downsizing may allow definitive treatment to be performed combining this approach with other oncologic treatments including surgery, systemic therapy and even conventional photon or particle therapy. You can find the text addressing this issue here: line 383-400 in the initial PDF manuscript (before this revision has been made). In my modest opinion, that is enough, I would prefer not to speculate too much, more than it has been already done.
Reviewer 2 Report
The paper investigates novel technologies (heavy ion / proton irradiation) and their induced non-targeted and remote effects on bulky unresectable tumours. The paper is timely and present an important topic given the increasing use of proton/heavy ion therapy on aggressive and hard-to-manage cancers. The authors present a rather unconventional planning and treatment delivery for a small cohort of 11 patients, for palliative intent.
The following are my comments that the authors should address for a better readability of the paper:
(1) Abstract: Please refrain from using abbreviations in the abstract which are not defined.
(2) Introduction: Both bystander and abscopal effects should be briefly described in this section.
(3) Introduction: The challenge posed by tumour hypoxia should be introduced / briefly discussed before the first mention of PATHY. Here the authors should specify the advantage of high LET radiation over low LET radiation in managing hypoxic subvolumes, as a justification for your clinical study with heavy ions (which they do later in the section).
(4) lines 80-82 – “It requires the use of an unconventional radiotherapy technique based on partial tumor irradiation as for example used in GRID or LATTICE whose radiobiology implicates the bystander effects” – and not only bystander effects but also immunological effects that might lead to superior tumour response when these techniques are employed:
https://pubmed.ncbi.nlm.nih.gov/32003574/
https://pubmed.ncbi.nlm.nih.gov/35328787/
(5) Methods: Since the authors emphasise in the Introduction the hypoxic subvolumes, it is not clear if there was any consideration of hypoxia when delineating the bystander tumour volume in the patient cohort. Please clarify in the manuscript.
Author Response
Thank you very much for all constructive comments and suggestions! We did take them all in consideration, point by point, in order to improve the quality of our paper:
“(1) Abstract: Please refrain from using abbreviations in the abstract which are not defined.”
Response: done! Thanks!
“(2) Introduction: Both bystander and abscopal effects should be briefly described in this section.”
Response: done! Thanks!
“(3) Introduction: The challenge posed by tumour hypoxia should be introduced / briefly discussed before the first mention of PATHY. Here the authors should specify the advantage of high LET radiation over low LET radiation in managing hypoxic subvolumes, as a justification for your clinical study with heavy ions (which they do later in the section).”
Response: done! Thanks!
“(4) lines 80-82 – “It requires the use of an unconventional radiotherapy technique based on partial tumor irradiation as for example used in GRID or LATTICE whose radiobiology implicates the bystander effects” – and not only bystander effects but also immunological effects that might lead to superior tumour response when these techniques are employed:
https://pubmed.ncbi.nlm.nih.gov/32003574/
https://pubmed.ncbi.nlm.nih.gov/35328787/”
Response: well, yes of course! If we said that those techniques implicates the bystander effect, thus, we thought that the bystander effect at least partially implies an immunogenic effect. Anyway, this was improved! Thanks!
“(5) Methods: Since the authors emphasise in the Introduction the hypoxic subvolumes, it is not clear if there was any consideration of hypoxia when delineating the bystander tumour volume in the patient cohort. Please clarify in the manuscript.”
Response: You´re right! We explained simply that the targeted “BTV” volume was delineated based on our previous experiences indicating that on average 30% (+/- 3%) centrally located tumor volume was relevant in terms of the tumor hypoxia, so that is why we have “chose” those 30% of the whole bulky tumor mass to be targeted now. In our ongoing prospective trial we actually do use the hypoxia-specific PET tracer for the target delineation, but we didn´t use any hypoxia-specific imaging for those 11 patients presented in this study. I thought that was clear, but obviously not enough. Now I added an additional explanation. Done! Thanks!
Reviewer 3 Report
Review „Novel Carbon-Ion and Proton Partial irradiation of recurrent unresectable bulky tumors (Particle-PATHY): Early indication of effectiveness and safety.”
Cancers
Thank you very much for giving me the opportunity to review this fantastic manuscript. The presented philosophy of radiotherapy, the corresponding radiotherapy planning and the clinical data are highly interesting and definitely worthy of publication.
However, the authors should address some points before publication:
Line 50 ff: “This is quite understandable given the very 50 large volume of these tumors and their very close relationship with neighboring critical organs, which do not allow the ablative radiation dose to be delivered by means of conventional radiotherapy.”
I suggest to rephrase: “This is understandable given the large volume of these tumors and their close relationship with neighboring critical organs, which do not allow the ablative radiation dose to be delivered through conventional radiotherapy.”
Line 58 – Reference 1 -> Please adapt the reference to the formatting requirements of the journal.
Line 63: this sentence is too long:
“Dealing with highly complex clinical scenarios encompassing the bulky tumors that are both unresectable and unsuitable for conventional radio-chemotherapy, in patients with high symptom burden and low performance status, this approach as an alternative to the standard palliative or BSC showed encouraging results in terms of immunogenicity, palliation, demonstrated that those results are likely caused immuno-modulatory effects which were previously explained by immunohistochemistry and gene-expression analyses and by the high rate of bystander and abscopal effects.”
I suggest to rephrase:
“Dealing with highly complex clinical scenarios encompassing the bulky tumors that are unresectable and unsuitable for conventional radio-chemotherapy in patients with high symptom burden and low-performance status, this approach is an alternative to the standard palliative or BSC. It showed encouraging results in terms of immunogenicity and palliation., demonstrated that those results are likely caused immuno-modulatory effects which were previously explained by immunohistochemistry and gene-expression analyses and by the high rate of bystander and abscopal effects.” -> The content of the last sentence is already elaborated below, so it can be omitted here.
Line 80, citation 6: This is quite an old citation, is anything younger available?
Line 81: what do “GRID” and “LATTICE” stand for? Citations?
Line 88 f: “Their specific physical and biological features make them ideally suited to achieve more efficiently the goals required in terms of immunomodulatory properties: neutralization of tumor hypoxia known as strong immunosuppressor as well as sparing the outer circumferences of tumor and normal tissue, i.e,. the PIM-volume (=Peritumoral 91 Immune Microenvironment).”
Please rephrase: “Their specific physical and biological features make them ideally suited to achieve more efficiently the goals required in terms of immunomodulatory properties: neutralization of tumor hypoxia, known as strong immunosuppressor, as well as sparing the outer circumferences of tumor and normal tissue, i.e., the PIM-volume (=Peritumoral Immune Microenvironment).”
Line 107: “…Review Board (Fig. 1).” Instead of “…Review Board. (Fig. 1).”
Line 168: 10-15Gy RBE?
Line 179 ff: all “Gy” are “Gy RBE”?
Line 201: Reference for CTCAE is missing
Line 212: “They were all pre-treated per standard of care treatments including the systemic therapy, surgery and radiotherapy resulting subsequently in progression disease (PD).”
Please rephrase: “They were all pretreated per standard of care treatments, including systemic therapy, surgery, and radiotherapy resulting subsequently in progressive disease (PD).”
Line 229, Table 2: first patient: what is the difference between “EBRT” and “RT”? Please specify in the commentary of the table, or omit "RT" in favour of "EBRT". See also patients 4,5,10
Line 302, Figure 6: to proof the abscopal effects, please provide a dose distribution of the distant lymph node metastases. In a horizontal section, the exclusion of the lymph node metastases from the radiotherapy should be easy to see.
Figure 7: same request as Figure 6: to proof the abscopal effects, please provide a dose distribution of the distant metastasis. In addition, please label the individual images and mention them in the caption.
Figure 8: In Figure 8, the interpretation of the regression of the metastasis marked in yellow as “abscopal” is questionable, as it was most likely in the beam path of the GTV. Therefore, please add the dose distribution here to assess an abscopal effect as more plausible.
Table 4B: DOWNSIZING/RETROSTAGING
I recommend deleting “Retrostaging”. This term is quite unusual.
Discussion
Line 361 “The scope of this study was not to assess the underlying physiologic mechanisms resulting in the favorable clinical outcomes;…” -> please change to: “The scope of this study was not to assess the underlying physiologic mechanisms resulting in the favorable clinical outcomes.”
Line 378 “Hence, conventional (“whole-tumor”) 378 photon-based radiotherapy is not optimally designed to be immuno-stimulative”.
This sentence cannot be left as it is: with photons, an ablative radiation dose of > 20 Gy can be delivered to an (optimally shaped) metastasis in one therapy session in the context of SABR, with an extremely steep dose fall-off to the surrounding tissues. Of course, these metastases are much smaller than the large tumors irradiated in their study. Please mention this technique in your discussion!
Furthermore, you must not and cannot attribute the observed effects solely to the immunological effects of radiotherapy, but must also take into account in the discussion that you gave 3 x high RT-doses, which can already develop a high tumouricidal effect in themselves. This very likely explains the local effects you observed. E.g.
- Garcia-Barros M, Paris F, Cordon-Cardo C, et al. (2003): Tumor Response to Radiotherapy Regulated by Endothelial Cell Apoptosis. Science 300:1155-1159
- Bodo S, Campagne C, Thin TH, et al. (2019): Single-dose radiotherapy disables tumorcell homologous recombination via ischemia/reperfusion injury. J Clin Invest. 129: 786-801
Furthermore, it should be discussed that not all tumours have the same peritumoral microenviroment (e.g. Tsujikawa T, Mitsuda J, Ogi H, Miyagawa-Hayashino A, Konishi E, Itoh K, Hirano S. Prognostic significance of spatial immune profiles in human solid cancers. Cancer Sci. 2020 Oct;111(10):3426-3434. doi: 10.1111/cas.14591. Epub 2020 Aug 12. PMID: 32726495; PMCID: PMC7540978). This could also be an explanation for the partially different response of the tumours.
Author Response
Thank you very much for all constructive comments and suggestions! We did take them all in consideration, point by point, in order to improve the quality of our paper:
“1) Line 50 ff: “This is quite understandable given the very 50 large volume of these tumors and their very close relationship with neighboring critical organs, which do not allow the ablative radiation dose to be delivered by means of conventional radiotherapy.”
I suggest to rephrase: “This is understandable given the large volume of these tumors and their close relationship with neighboring critical organs, which do not allow the ablative radiation dose to be delivered through conventional radiotherapy.”
Response: done! Thanks!
“2) Line 58 – Reference 1 -> Please adapt the reference to the formatting requirements of the journal.”
Response: done! Thanks!
“3) Line 63: this sentence is too long:
“Dealing with highly complex clinical scenarios encompassing the bulky tumors that are both unresectable and unsuitable for conventional radio-chemotherapy, in patients with high symptom burden and low performance status, this approach as an alternative to the standard palliative or BSC showed encouraging results in terms of immunogenicity, palliation, demonstrated that those results are likely caused immuno-modulatory effects which were previously explained by immunohistochemistry and gene-expression analyses and by the high rate of bystander and abscopal effects.”
I suggest to rephrase:
“Dealing with highly complex clinical scenarios encompassing the bulky tumors that are unresectable and unsuitable for conventional radio-chemotherapy in patients with high symptom burden and low-performance status, this approach is an alternative to the standard palliative or BSC. It showed encouraging results in terms of immunogenicity and palliation., demonstrated that those results are likely caused immuno-modulatory effects which were previously explained by immunohistochemistry and gene-expression analyses and by the high rate of bystander and abscopal effects.” -> The content of the last sentence is already elaborated below, so it can be omitted here.”
Response: done! Thanks!
“4) Line 80, citation 6: This is quite an old citation, is anything younger available?”
Response: yes, it is available of course, but I think that paper although a bit older is of a very good quality, probably better than many newer! I appreciate your suggestion, but if you don´t mind I would keep that one. Thanks!
“5) Line 81: what do “GRID” and “LATTICE” stand for? Citations?”
Response: GRID and LATTICE are not any kind of abbreviations but the names of unconventional radiation techniques. The citations you have suggested have been added! Thanks!
“6) Line 88 f: “Their specific physical and biological features make them ideally suited to achieve more efficiently the goals required in terms of immunomodulatory properties: neutralization of tumor hypoxia known as strong immunosuppressor as well as sparing the outer circumferences of tumor and normal tissue, i.e,. the PIM-volume (=Peritumoral 91 Immune Microenvironment).”
Please rephrase: “Their specific physical and biological features make them ideally suited to achieve more efficiently the goals required in terms of immunomodulatory properties: neutralization of tumor hypoxia, known as strong immunosuppressor, as well as sparing the outer circumferences of tumor and normal tissue, i.e., the PIM-volume (=Peritumoral Immune Microenvironment).”
Response: sure! Done! Thanks!
“7) Line 107: “…Review Board (Fig. 1).” Instead of “…Review Board. (Fig. 1).”
Response: Done! Thanks!
“8) Line 168: 10-15Gy RBE?
“9) Line 179 ff: all “Gy” are “Gy RBE”?”
Response: well, we reported in the methods, under “dose prescription” (line 187 in the initial PDF manuscript before revision) that the reported doses in Gy are RBE-weighted doses calculated from the physical dose using the LEM model for carbon-ions while for protons a fixed RBE of 1.1 was assumed. I have improved few locations in the text where was reported in a different way to meet the consistency! Thanks!
“10) Line 201: Reference for CTCAE is missing”
Response: right! Sorry for that! Added! Thanks!
“11) Line 212: “They were all pre-treated per standard of care treatments including the systemic therapy, surgery and radiotherapy resulting subsequently in progression disease (PD).”
Please rephrase: “They were all pretreated per standard of care treatments, including systemic therapy, surgery, and radiotherapy resulting subsequently in progressive disease (PD).”
Response: sure! Thanks!
“12) Line 229, Table 2: first patient: what is the difference between “EBRT” and “RT”? Please specify in the commentary of the table, or omit "RT" in favour of "EBRT". See also patients 4,5,10”
Response: Right! Done! Thanks!
“13-15) Line 302, Figure 6: to proof the abscopal effects, please provide a dose distribution of the distant lymph node metastases. In a horizontal section, the exclusion of the lymph node metastases from the radiotherapy should be easy to see.
Figure 7: same request as Figure 6: to proof the abscopal effects, please provide a dose distribution of the distant metastasis. In addition, please label the individual images and mention them in the caption.
Figure 8: In Figure 8, the interpretation of the regression of the metastasis marked in yellow as “abscopal” is questionable, as it was most likely in the beam path of the GTV. Therefore, please add the dose distribution here to assess an abscopal effect as more plausible.”
Response: very good point! Thanks! I created a new figure 9 because those 6-8 are already very large and there was no space enough for me to add an additional component, especially because the numbers indicating the radiation doses at the level of the abscopal tumor sites are small.
“16) Table 4B: DOWNSIZING/RETROSTAGING
I recommend deleting “Retrostaging”. This term is quite unusual.”
Response: You saw that all the bulky tumors had as for the stage “T4” indicating a maximal tumor extension and local invasiveness. Actually, following the PARTICLE-PATHY many of them moved to the lower T1-T3 stage (thus, were retro-staged). However, it´s not that important, downsizing is enough to make a point. It was deleted. Thanks!
“17) Line 361 “The scope of this study was not to assess the underlying physiologic mechanisms resulting in the favorable clinical outcomes;…” -> please change to: “The scope of this study was not to assess the underlying physiologic mechanisms resulting in the favorable clinical outcomes.”
Response: done! Thanks!
“18) Line 378 “Hence, conventional (“whole-tumor”) 378 photon-based radiotherapy is not optimally designed to be immuno-stimulative”. This sentence cannot be left as it is: with photons, an ablative radiation dose of > 20 Gy can be delivered to an (optimally shaped) metastasis in one therapy session in the context of SABR, with an extremely steep dose fall-off to the surrounding tissues. Of course, these metastases are much smaller than the large tumors irradiated in their study. Please mention this technique in your discussion!
Response: I disagree with your statement. At least, I believe there is a misunderstanding when the text in the line 378 is considered, because I don´t see much connection with what you explained above. Let me explain: my point was that the conventional radiotherapy was not designed to be immuno-stimulative, because it kills the immune system cells within the CTV and PTV regions (where you have the normal tissue containing tumor-infiltrating and resident immune cells). My point is that if we want the radiation-immunogenic effects, those immune cells have to be spared. This applies also to the SBRT (because of the whole-tumor irradiation) no matter how steep dose fall-off is, because it will always deliver at the tumor surface and surrounding healthy tissues within the PTV enough high radiation dose to kill the radio-sensitive immune system cells necessary to mediate for the non-targeted effects. Everything is fine with the photon-radiotherapy of course! It´s just the form in which is given is not in favor of immunogenicity. The same is true also for the particle therapy when is given as the traditional whole-tumor irradiation. Thank you!
“19) Furthermore, you must not and cannot attribute the observed effects solely to the immunological effects of radiotherapy, but must also take into account in the discussion that you gave 3 x high RT-doses, which can already develop a high tumouricidal effect in themselves. This very likely explains the local effects you observed. E.g.
- Garcia-Barros M, Paris F, Cordon-Cardo C, et al. (2003): Tumor Response to Radiotherapy Regulated by Endothelial Cell Apoptosis. Science 300:1155-1159
- Bodo S, Campagne C, Thin TH, et al. (2019): Single-dose radiotherapy disables tumorcell homologous recombination via ischemia/reperfusion injury. J Clin Invest. 129: 786-801”
Response: I do not attribute the observed effects solely to the immunological effects of radiotherapy! See the text in line 58-59 saying that “This technique was purposefully designed to add to the radiation-mediated tumor cell killing also the immune-mediated killing component.” It is true and I agree with you that a significant tumor portion will die due to the direct radiation tumoricidal effect, no question about it! However, definitely, there is an immune-mediated tumoricidal component that boosts the observed radiation anti-tumor effects, and this is not just the hypothesis but proven contention: 1.) because in a previous study the immunohistochemistry and gene-expression findings of the partially irradiated and distant unirradiated tumor sites showed immune- and cytokine-mediated tumor cell killing (https://www.mdpi.com/2072-6694/13/1/50); 2.) Because of the presence of abscopal effects (regression of unirradiated tumors); 3.) Because on average “only” 29.6% of the tumor mass was exposed to that high radiation dose while the peripheral tumor portion received radiation dose similar to that of the PIM (see IMMUNE-SPARING: PIM-DOSE in table 3: Dmean 5.8Gy in 3 fractions or D30%<0.7Gy in 3 fractions), how then such treatment can induce a complete tumor response observed in this but also in previous series (https://www.mdpi.com/2072-6694/13/1/50)? Finally, concerning the references you proposed to explain the observed anti-tumor effects (Garcia-Barros M et al./ Bodo S. et al.), you probably didn´t realized that approximately 90-95% of the targeted BTV-volume consisted of (peri)necrotic tumor “tissue” which does not contain functional tumor vessels that could be damaged and therefore responsible for anti-tumor effects. Again, at least the majority if not all of the tumor vessels are located in tumor´s peripheral part that was intentionally spared from irradiation/non-targeted with ablative radiation dose. Even if I would ignore the above mentioned arguments, it is hard for me to believe in such kind of hypothesis you proposed because it fails to explain the induced abscopal effects.
“20) Furthermore, it should be discussed that not all tumours have the same peritumoral microenviroment (e.g. Tsujikawa T, Mitsuda J, Ogi H, Miyagawa-Hayashino A, Konishi E, Itoh K, Hirano S. Prognostic significance of spatial immune profiles in human solid cancers. Cancer Sci. 2020 Oct;111(10):3426-3434. doi: 10.1111/cas.14591. Epub 2020 Aug 12. PMID: 32726495; PMCID: PMC7540978). This could also be an explanation for the partially different response of the tumours.”
Response: This I would simply leave for discussion when my new data on that topic will soon be available…Currently, I think it would be too much of speculation…This was a small study showing some early but interesting and indicative results; However, I don't have any relevant findings to support the discussion about the types of the microenvironment…I hope you understand and accept my opinion on this. Thank you!